# Bridging Text and Knowledge by Learning Multi-Prototype Entity Mention Embedding

## Abstract

Integrating text and knowledge into a unified semantic space has attracted significant research interests recently. However, the ambiguity in the common space remains a challenge, namely that the same mention phrase usually refer to various entities. In this paper, to deal with the ambiguity of entity mentions, we propose a novel Multi-Prototype Mention Embedding model, which learns multiple sense embeddings for each mention by jointly modeling words from textual contexts and entities derived from a knowledge base. In addition, we further design an efficient language model based approach to disambiguate each mention to a specific sense. In experiments, both qualitative and quantitative analysis demonstrate the high quality of the word, entity and multi-prototype mention embeddings. Using entity linking as a study case, we apply our disambiguation method as well as the multi-prototype mention embeddings on the benchmark dataset, and achieve the state-of-the-art.

## 1 Introduction

Jointly learning text and knowledge representations in a unified vector space greatly benefits many Natural Language Processing (NLP) tasks, such as knowledge graph completion (Han et al., 2016; Wang and Li, 2016), relation extraction (Weston et al., 2013), word sense disambiguation (Mancini et al., 2016) and entity linking (Huang et al., 2015).

Existing work can be roughly divided into two categories. One is encoding words and entities into a unified vector space using Deep Neural Networks (DNN). These methods suffer from the problem of expensive training and have great limitations on the size of word and entity vocabulary (Han et al., 2016; Toutanova et al., 2015; Wu et al., 2016). The other is to learn word and entity embeddings separately, and then align similar words and entities into a common space with the help of Wikipedia hyperlinks, so that they share similar representations (Wang et al., 2014; Yamada et al., 2016).

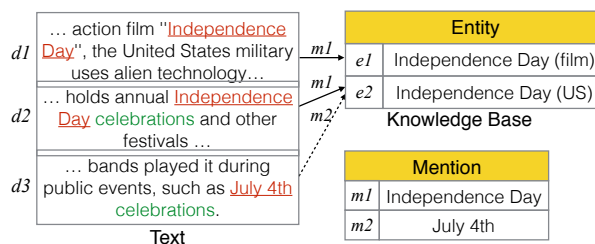

Figure 1: Examples.

However, there are two major problems arising from directly integrating word and entity embeddings into a unified semantic space. First, mention phrases are of great ambiguity and can refer to multiple entities in the common space. As shown in Figure 1, the same mention *independence day* ($m_1$) can either refer to a holiday: *Independence Day (US)* or a film: *Independence Day (film)*. Second, an entity often has various aliases when mentioned in various contexts, which implies a much larger size of mention vocabulary compared with entities. For example, in Figure 1, the document $d_2$ and $d_3$ describes the same entity *Independence Day (US)* ($e_2$) with distinct mentions: *independence day* and *July 4th*. We observe a tens of millions of mention vocabulary on 5 millions of entities in Wikipedia.

To address these issues, we propose to learn multiple embeddings for each mention. Each embedding denotes a meaning, namely **mention sense**. Similar as the Word Sense Disambiguation

(WSD) problem (Reisinger and Mooney, 2010; Huang et al., 2012; Tian et al., 2014; Neelakan­tan et al., 2014; Li and Jurafsky, 2015), each men­tion sense can be disambiguated by textual con­text information as well as its reference entity. For example, in Figure 1, the *Independence Day* in $d_2$ shares similar contexts with the *July 4th* in $d_3$ and refers to the same entity. In contract, it may also refer to another entity when occur in distinct contexts, such as $d_1$. Further, to reduce the size of mention vocabulary, we assume that mentions share the same sense representation if they refer to the same entity, e.g., the *Independence Day* in $d_2$ and *July 4th* in $d_3$ will have the same mention rep­resentation as they refer to the same holiday. Thus, mention senses can serve as a repository and each mention can be mapped to them by a pre-defined dictionary.

In this paper, we propose a novel **Multi­Prototype Mention Embedding (MPME)** model, which jointly learns the representations of words, entities, and mentions at sense level. The basic idea behind it is to use both textual context in­formation and knowledge of reference entity to distinguish different mention senses. Following the frameworks in (Wang et al., 2014; Yamada et al., 2016), we use separate models to learn the representations for words, entities and men­tions, and further align them by a unified optimiza­tion objective. Extending from skip-gram model and CBOW model, our model can be trained effi­ciently (Mikolov et al., 2013a,b) even on a large scale corpus. In addition, we also design an language model based approach to determine the sense for each mention in a document based on multi-prototype mention embeddings.

For evaluation, we first provide qualitative anal­ysis to verify the effectiveness of MPME to bridge text and knowledge representations at the sense level. Then, separate tasks for words and entities show improvements by incorporating our word, entity and mention representations. Finally, using entity linking as a study case, experimental results on the benchmark dataset demonstrate the effec­tiveness of our embedding model as well as the disambiguation method.

## 2 Preliminaries

In this section, we formally define the input and output of multi-prototype mention embedding.

A **knowledge base** $\mathcal{KB}$ contains a set of entities $\mathcal{E} = \{e_j\}$, and their relations. Following (Yamada et al., 2016), we use Wikipedia as the given knowl­edge base, and organize it as a directed network, namely knowledge network: nodes denote enti­ties, and edges are outlinks from wikipedia pages. We define the neighbors of $e_j$ on the network as $\mathcal{N}(e_j)$.

A **text corpus** $\mathcal{D}$ is a set of sequential words $\mathcal{D} = \{w_1, \cdots, w_i, \cdots, w_{|\mathcal{D}|}\}$, where $w_i$ is the $i$th word and $|\mathcal{D}|$ is the length of the word sequence. We use $m_l$ to denote an entity mention (perhaps consisting of multiple words). We define an anno­tated text corpus as $\mathcal{D}' = \{x_1, \cdots, x_i, \cdots, x_{|\mathcal{D}'|}\}$, where $x_i$ corresponds to a word $w_i$ or a mention $m_l$. We define the words around $x_i$ within a pre­defined window as its context words $\mathcal{C}(\cdot)$.

An **Anchor** indicates the Wikipedia hyperlinks from mention $m_l$ linking to entity $e_j$, and is repre­sented as a pair $< m_h, e_j > \in \mathcal{A}$. The anchors pro­vide mention boundaries as well as their reference entities from Wikipedia articles. These Wikipedia articles are used as annotated text corpus $\mathcal{D}'$ in this paper.

**Multi-Prototype Mention Embedding** . Given a $\mathcal{KB}$, an annotated text corpus $\mathcal{D}'$ and a set of anchors $\mathcal{A}$, multi-prototype mention embedding is to learn multiple sense embeddings $\mathbf{s_j}^l \in \mathbb{R}^k$ for each mention $m_l$ as well as word embeddings $\mathbf{w}$ and entity embeddings $\mathbf{e}$. We use $\mathcal{M}_l^* = \{s_j^l\}$ to denote the sense set of mention $m_l$, where each $s_j^l$ refers to an entity $e_j$. Thus, the vocabulary size is reduced to a fixed number $|\{s_j^*\}| = |\mathcal{E}|$. We use $s_j^*$ to denote the shared sense of mentions referring to entity $e_j$.

**Example** As shown in Figure 1, *Independence Day* $(m_1)$ is learned with two mention senses $s_1^1, s_2^1$, and *July 4th* $(m_2)$ has one mention sense $s_2^2$. Based on the assumption in Section 1, we have $s_2^* = s_2^1 = s_2^2$ referring to entity *Independence Day (US)* $(e_2)$.

## 3 An Overview of Our Method

Given knowledge base $\mathcal{KB}$, annotated text corpus $\mathcal{D}'$ and a set of anchors $\mathcal{A}$, we aim to jointly learn word, entity and mention sense representations: $\mathbf{w}, \mathbf{e}, \mathbf{s}$.

As shown in Figure 2, our framework contains two key components:

**Mention Sense Mapping** To reduce the mention vocabulary, each mention is mapped to a set of shared mention senses according to a predefined

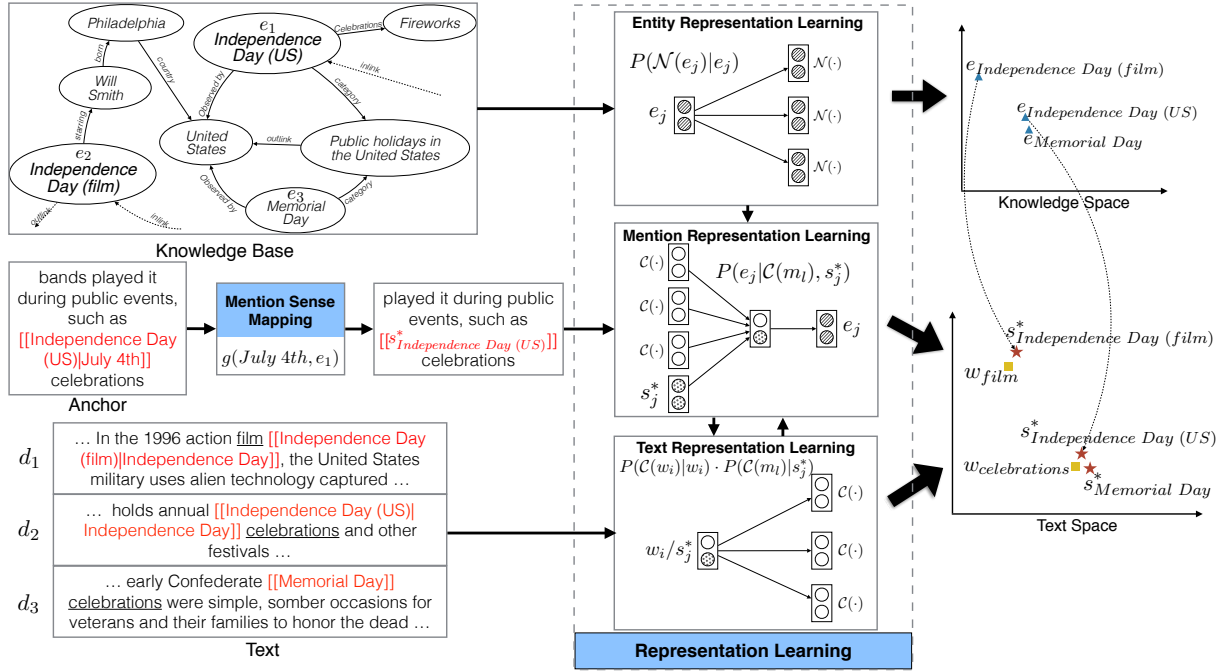

Figure 2: Framework of Multi-Prototype Mention Embedding model.

dictionary. Similar to (Shen et al., 2015), we build the dictionary by collecting mention-entity pairs $< m_l, e_j >$ from Wikipedia page titles, redirect pages and anchors. According to $e_j$, each pair provides a candidate mention sense for $m_l$, so we formally define the mapping as: $\mathcal{M}_l^* = \bigcup g(< m_l, e_j >) = \{s_j^*\}$, where $g(\cdot)$ denotes the mapping function from entity mention to its reference entity, as well as the mapping from each mention to its sense.

We directly use the anchors contained in the annotated text corpus $D'$ for training. As Figure 2 shows, we replace the anchor <*July 4th, Independence Day (US)*> with the corresponding mention sense: *Independence Day (US)*.

**Representation Learning** Using $\mathcal{KB}$, $\mathcal{A}$ and $\mathcal{D}'$ as input, we design three separate models and a unified optimization objective to jointly learn entity, word and mention sense representations into two semantic spaces. Entity embeddings can reflect their relatedness in the network, e.g. *Independence Day (US)* ($e_1$) and *Memorial Day* ($e_3$) are close to each other because they share some common neighbors, e.g., *United States* and *Public holidays in the United States*, as shown in Figure 2.

Word and mention embeddings are learned from the same semantic space. As two basic units in $\mathcal{D}'$, their embeddings represent their distributed semantics in texts. For example, mention *Inde-*

*pendence Day* and word *celebrations* co-occur frequently when it refers to the holiday: *Independence Day (US)*, thus they have similar representations. Without disambiguating the mention senses, some words, e.g., *film*, will also share similar representations as *Independence Day*.

Besides, by introducing entity embeddings into our MPME framework, the knowledge information will also be distilled into mention sense embeddings, so that the mention sense *Memorial Day* can be similar as *Independence Day (US)*.

**Mention Sense Disambiguation** Given a document, in order to disambiguate the sense for each mention, we design a language model based approach to measure the similarity of each candidate mention sense with the local contexts as well as the neighbor mentions occurring in a limited size of window.

## 4 Representation Learning

### 4.1 Skip-Gram and CBOW model

Skip-gram and CBOW models (Mikolov et al., 2013a,b) are widely used to learn distributed word representations. Given a sequence of words $\mathcal{D}$, the optimization objective of Skip-gram model is to maximize the average log probability:

$$\mathcal{L} = \sum_{w_i \in \mathcal{D}} \sum_{w_o \in \mathcal{C}(w_i)} P(w_o|w_i) \quad (1)$$

In contrast, CBOW model aims to predict the current word given its context words:

$$\mathcal{L} = \sum_{w_i \in \mathcal{D}} P(w_i | \mathcal{C}(w_i)) \qquad (2)$$

Formally, the conditional probability $P(w_o|w_i)$ is defined using a softmax function:

$$P(w_o|w_i) = \frac{\exp(\mathbf{w_i} \cdot \mathbf{w_o})}{\sum_{w_o \in \mathcal{D}} \exp(\mathbf{w_i} \cdot \mathbf{w_o})} \qquad (3)$$

where $\mathbf{w_i}$, $\mathbf{w_o}$ denote the input and output word vectors during training. Furthermore, these two models can be accelerated by using hierarchical softmax or negative sampling (Mikolov et al., 2013a,b).

### 4.2 Entity Representation Learning

Given a knowledge base $\mathcal{KB}$, we aim to learn entity embeddings by modeling "contextual" entities, so that the entities sharing more common neighbors tend to have similar representations. Therefore, we extend Skip-gram model to a network by maximizing the log probability of being a neighbor entity.

$$\mathcal{L}_e = \sum_{e_j \in \mathcal{E}} \log P(\mathcal{N}(e_j)|e_j) \qquad (4)$$

Clearly, the neighbor entities serve a similar role as the context words in Skip-gram model. As shown in Figure 2, entity *Memorial Day* ($e_3$) also has the neighbors of *United States* and *Public holidays in the United States*, thus their embeddings are close in knowledge space. These entity embeddings will be later used to learn mention representations.

### 4.3 Mention Representation Learning

As mentioned above, the textual context information and reference entity are helpful to distinguish different senses for a mention. Thus, given an anchor $< m_l, e_j >$ and its context words $\mathcal{C}(m_l)$, we combine mention sense embeddings with its context word embeddings to predict the reference entity by extending CBOW model. The objective function is as follows:

$$\mathcal{L}_m = \sum_{<m_l,e_j> \in \mathcal{A}} \log P(e_j | \mathcal{C}(m_l), s_j^*) \qquad (5)$$

where $s_j^* = g(< m_l, e_j >)$. Thus, if two mentions refer to similar entities and share similar contexts, they tend to be close in semantic vector space. Take Figure 1 as an example again, mention *Independence Day* and *Memorial Day* refer to similar entities *Independence Day (US)* ($e_1$) and *Memorial Day* ($e_2$) and share some similar context words, e.g., *celebrations* in documents $d_2, d_3$, so their sense embeddings are close to each other in text space.

### 4.4 Text Representation Learning

Instead of directly using a word or a mention to predict the context words, we incorporate mention sense to joint optimize word and sense representations, which can avoid some noise introduced by the multiple senses mentions. For example, in Figure 2, without identifying the mention *Independence Day* as the holiday or the film, various dissimilar context words such as *celebrations* and *film* in documents $d_1, d_2$ will share similar semantics, which will further affect the performance of entity representations during joint training.

Given the annotated corpus $\mathcal{D}'$, we use a word $w_i$ or a mention sense $s_j^*$ to predict the context words by maximizing the following objective function:

$$\mathcal{L}_w = \sum_{w_i, m_l \in \mathcal{D}'} \log P(\mathcal{C}(w_i)|w_i)$$
$$+ \log P(\mathcal{C}(m_l)|s_j^*) \qquad (6)$$

where $s_j^* = g(< m_l, e_j >)$ is obtained from anchors in Wikipedia articles.

Thus, words and mention senses will share the same vector space, where similar words and mention senses are close to each other, such as *celebrations* and *Independence Day (US)* because they frequently occur in the same contexts.

Similar to WDS, we maintain a context cluster for each mention sense, which can be used for mention sense disambiguation (Section 5). The context cluster of a mention sense $s_j^*$ contains all the context vectors of its mention $m_l$. We compute context vector of $m_l$ by averaging the sum of its context word embeddings: $\frac{1}{|\mathcal{C}(m_l)|} \sum_{w_j \in \mathcal{C}(m_l)} \mathbf{w_j}$. Further, the center of a context cluster $\mu_j^*$ is defined as the average of context vectors of all mentions which refer to the sense. These context clusters will be later used to disambiguate the sense of a given mention with its contexts.

## 4.5 Joint Training

Considering all the above representation learning components, we define the overall objective function as linear combinations:

$$\mathcal{L} = \mathcal{L}_w + \mathcal{L}_e + \mathcal{L}_m \qquad (7)$$

The training of MPME is to maximize the above function, and iteratively update three types of embeddings. Also, we use negative sampling technique for efficiency (Mikolov et al., 2013a).

## 5 Mention Sense Disambiguation

As we have learned multiple sense representations for each mention, given a new mention with its context, we also need to disambiguate and link the mention to a pre-learned sense.

Given an annotated document $\mathcal{D}'$ including $\mathcal{M}$ mentions, we first map each mention $m_l \in \mathcal{M}$ to a set of senses according to Section 3: $\mathcal{M}_l^* = \{s_j^* | s_j^* \in g(m_l)\}$. Then, based on language model, we identify the correct sense by maximizing a joint probability of all mention senses contained in the document. However, the global optimum is expensive with a time complexity of $O(|\mathcal{M}||\mathcal{M}_l^*|)$. Thus, we approximately identify each mention sense independently:

$$
\begin{aligned}
&P(\mathcal{D}', \dots, s_j^*, \dots, ) \\
&\approx \prod P(\mathcal{D}'|s_j^*) \cdot P(s_j^*) \qquad (8) \\
&\approx \prod P(\mathcal{C}(m_l)|s_j^*) \cdot P(\hat{\mathcal{N}}(m_l)|s_j^*) \cdot P(s_j^*)
\end{aligned}
$$

where $P(\mathcal{C}(m_l)|s_j^*)$ denotes the probability of the local contexts of $m_l$ given its mention sense $s_j^*$, namely local similarity. we define it proportional to the cosine similarity between current context vector and the sense context cluster center $\mu_j^*$ as described in Section 4.4. It measures how likely a mention sense occurring together with current context words. For example, given the mention sense *Independence Day (film)*, word *film* is more likely to appear within the context than the word *celebrations*.

$P(\hat{\mathcal{N}}(m_l)|s_j^l)$ denotes the probability of the contextual mentions of $m_l$ given its sense $s_j^l$, namely global probability, where $\hat{\mathcal{N}}(m_l)$ is the collection of the neighbor mentions occurring together with $m_l$ in a predefined context window. We define it proportional to the cosine similarity between mention sense embeddings and the neighbor mention vector, which is computed similar to context vector: $\sum \frac{1}{|\hat{\mathcal{N}}(m_l)|} \hat{s}_j^l$, where $\hat{s}_j^l$ is the correct sense for $m_l$. The correct sense can be induced using either L2R (left to right) or S2C (simple to complex) orders (Chen et al., 2014).

Global probability assumes that there should be consistent semantics in a context window, and measures whether all neighbor mentions are related. For instance, two mentions *Memorial Day* and *Independence Day* occurring in the same document. If we already know that *Memorial Day* denotes a holiday, then obviously *Independence Day* has higher probability of being the holiday than the film.

$P(s_j^*)$ is a prior probability of sense $s_j^*$ indicating how possible it occurs without considering any additional information. We define it proportional to the frequency of sense $s_j^*$ in Wikipedia anchors:

$$ P(s_j^*) = (\frac{|\mathcal{A}_{s_j^*}|}{|\mathcal{A}|})^\gamma \quad \gamma \in [0, 1] $$

where $\mathcal{A}_{s_j^*}$ is the set of anchors annotated with $s_j^*$, and $\gamma$ is a smoothing hyper-parameter to control the impact of prior on the overall probability, which is set by experiments (Section 6.4.2).

## 6 Experiments

**Setup** We choose Wikipedia, the March 2016 dump, as training corpus, which contains nearly 75 millions of anchors, 180 millions of edges among entities and 1.8 billions of tokens after pre-processing. We then learn representations for 1.5 millions of words, 5 millions of entities and 1.7 millions of mentions. The entire training process in 10 iterations costs nearly 8 hours on the server with 64 core cpu and 188GB memory.

We use the default settings in word2vec[1], and set our embedding dimension as 200 and context window size as 5. For each positive example, we sample 5 negative examples.

**Baseline Methods** As far as we know, this is the first work to deal with mention ambiguity in the integration of text and knowledge representations, so there is no exact baselines for comparison. We use the method in (Yamada et al., 2016) as a baseline, marked as **ALIGN** because (1) this is the most similar work that directly aligns word and entity embeddings. (2) it achieves the state-of-the-art performance in entity linking task.

---

[1]https://code.google.com/archive/p/word2vec/

| | Mention Sense | Nearest words | Nearest entities |
|---|---|---|---|
| SPME | Independence Day | lee-jackson, thanksgiving, diwali, strassenfest, chiraghan | National Aboriginal and Torres Strait Islander Education Policy, E. Chandrasekharan Nair, Jean Aileen Little, Thessalian barbel, 1825 in birding and ornithology |
| MPME | Independence Day (US) | thanksgiving, parades, lee-jackson, festivities, celebrations | Memorial Day, Labor Day, Thanksgiving, Thanksgiving (United States), Saint Patrick's Day |
| | Independence Day (film) | robocop, clockstoppers, mindhunters, tarantino, terminator | The Terminator, True Lies, Total Recall (1990 film), RoboCop 2, Die Hard |

Table 1: The nearest neighbors of mention *Independence Day*.

To investigate the effect of multi prototype, we degrade our method to single prototype, which means to use one sense to represent all mentions with the same phrase, namely **S**ingle-**P**rototype **M**ention **E**mbedding (SPME). For example, we only use one unique sense vector for *Independence Day* whatever it denotes the holiday or the film.

## 6.1 Qualitative Analysis

We use cosine similarity to measure the similarity of two vectors, and present the top 5 nearest words and entities for two most popular senses of the mention *Independence Day*. Because ALIGN is incapable of dealing with multiple words, we only give the results of SPME and MPME.

As shown in Figure 1, without considering mention sense, the mention *Independence Day* can only show an dominant *holiday* sense based on SPME and ignore all other senses. Instead, MPME successfully learns two clear and distinct senses. For the sense *Independence Day (US)*, all of it nearest words and entities, such as *parades*, *celebrations*, and *Memorial Day*, are holiday related, while for another sense *Independence Day (film)*, its nearest words and entities, such as *robocop* and *The Terminator*, are all science fiction films. All the results demonstrate the effectiveness of our framework in multi-prototype mention embedding learning.

## 6.2 Entity Relatedness

To evaluate the quality of entity embeddings, we conduct experiments using the dataset which is designed for measuring entity relatedness (Ceccarelli et al., 2013; Huang et al., 2015; Yamada et al., 2016). The dataset contains 3314 entities, and each mention has 91 candidate entities on average with gold-standard labels indicating whether they are semantic related.

We compute cosine similarity between entity embeddings to measure their relatedness, and rank them in a descending order. To evaluate the ranking quality, we use two standard metrics:

normalized discounted cumulative gain (NDCG) (Järvelin and Kekäläinen, 2002) and mean average precision (MAP) (Schütze, 2008).

We design another baseline method: **Entity2vec**, which learns entity embeddings using the method described in Section 4.2, without joint training with word and mention sense embeddings.

Table 2: Entity Relatedness.

| | NDCG@1 | NDCG@5 | NDCG@10 | MAP |
|---|---|---|---|---|
| ALIGN | 0.416 | 0.432 | 0.472 | 0.410 |
| Entity2vec | 0.593 | 0.595 | 0.636 | 0.566 |
| SPME | 0.593 | 0.594 | 0.636 | 0.566 |
| MPME | **0.613** | **0.613** | **0.654** | **0.582** |

As shown in Table 2, ALIGN gets lower performance[2] than Entity2vec, because it doesn't consider the mention phrase ambiguity and yields lots of noise when forcing entity embeddings to satisfy word embeddings and aligning them into the unified space. For example, entity *Gente (magazine)* should be more relevance to entity *France*, the place where its company locates. However, ALIGN mixed various meanings of mention *Gente* (e.g. the song) and ranked some bands higher (e.g. entity *Poolside (band)*).

SPME also doesn't consider the ambiguity of mentions but achieves comparative results with Entity2vec. We analyze the reasons and find that, it can avoid some noise by using word embeddings to predict entities. MPME outperforms all the other methods, which demonstrates that the unambiguous textual information is helpful to refine the entity embeddings.

## 6.3 Word Analogical Reasoning

Following (Mikolov et al., 2013a; Wang et al., 2014), we use word analogical reasoning task to evaluate the quality of word embeddings. The dataset consists of 8869 semantic questions

---

[2]We failed to reproduce the positive result in the original paper, meanwhile the authors are unable to release their code.

(*"Paris":"France"::"Rome":?*), and 10675 syntactic questions (e.g. *"sit":"sitting"::"walk":?*). We solve it by finding the closest word vector $\mathbf{w}_?$ to $\mathbf{w}_{France} - \mathbf{w}_{Paris} + \mathbf{w}_{Rome}$ according to cosine similarity. We compute accuracy for top 1 nearest word to measure the performance.

Table 3: Word Similarity.

|  | Word2vec | ALIGN | SPME | MPME |
|---|---|---|---|---|
| Semantic | 66.78 | 68.34 | 71.65 | 71.65 |
| Syntactic | 61.58 | 59.73 | 55.28 | 54.75 |

We also adopt Word2vec[3] as an additional baseline method, which provides a standard to measure the impact from other components on word embeddings.

Table 3 shows the results. We can see that ALIGN, SPME and MPME, achieve higher performance in semantic task, because relations among entities (e.g. country-capital relation for entity *France* and *Paris*) enhance the semantics in word embeddings through jointly training. On the other hand, their syntactic performance decrease. We analyze the results and find that, most of errors are caused by the bias of semantics. For example, given a query *"pleasant":"unpleasant"::"possibly":?*, our model tends to return the word which has high semantic similarity with query words, such as *probably*, instead of the syntactical similar words, e.g., *impossibly*.

The word embeddings of MPME achieve the best semantic performance even be forced to satisfy with entity embeddings in joint training, mainly because (1) text representation learning has better generalization ability due to the larger size of training examples than entities' (i.e. 1.8b v.s. 0.18b) as well as relatively smaller size of vocabulary (i.e. 1.5m v.s. 5m). (2) unambiguous mention embeddings capture both textual context information and knowledge, and thus enhance word and entity embeddings.

## 6.4 A Case Study: Entity Linking

As a study case, we further apply the multi-prototype mention embeddings to entity linking task. Given mentions in text, entity linking aims to link them to a predefined knowledge base. One of the main challenges in this task is the ambiguity of entity mentions.

---

[3] https://code.google.com/archive/p/word2vec/

We use the public dataset AIDA created by (Hoffart et al., 2011), which includes 1393 documents and 27,816 mentions referring to Wikipedia entries. The dataset has been divided into 946, 216 and 231 documents for the purpose of training, developing and testing. Following (Pershina et al., 2015; Yamada et al., 2016), we use a public available dictionary to generate candidate entities and mention senses. For evaluation, we rank the candidate entities for each mention and report both standard micro (aggregates over all mentions) and macro (aggregates over all documents) precision over top-ranked entities.

### 6.4.1 Supervised Entity Linking

Yamada et al. (2016) designs a list of features for each mention and candidate entity pair. By incorporating these features into a supervised learning-to-rank algorithm, Gradient Boosting Regression Tree (GBRT), each pair is obtained a relevance score indicating whether they should be linked to each other. Following their recommended parameters, we set the number of trees as 10,000, the learning rate as 0.02 and the maximum depth of the decision tree as 4.

Based on word and entity embeddings learned by ALIGN, the key features in (Yamada et al., 2016) are from two aspects: (1) the cosine similarity between context words and candidate entity, and (2) the coherence among "contextual" entities in the same document.

To evaluate the performance of multi-prototype mention embeddings, we incorporate the following features into GBDT for comparison: (1) the cosine similarity between the current context vector and the sense context cluster center $\mu_j^*$, which denotes how likely the mention sense to refer to the candidate entity, (2) the cosine similarity between the current context vector and the mention sense embeddings.

Table 4: Performance of Supervised Method

|  | ALIGN | SPME | MPME |
|---|---|---|---|
| Micro P@1 | 0.828 | 0.820 | **0.851** |
| Macro P@1 | 0.862 | 0.844 | **0.881** |

As shown in Table 4, we can see that ALIGN performs better than SPME. This is because SPME learns word embeddings and entity embeddings in separate semantic spaces, and fails to measure the similarity between context words and candidate entities. However, MPME com-

Table 5: Performance of unsupervised methods

|  | Cucerzan | Kulkarni | Hoffart | Shirakawa | Alhelbawy | MPME (L2R) | MPME (S2C) |
|---|---|---|---|---|---|---|---|
| Micro P@1 | 0.510 | 0.729 | 0.818 | 0.823 | 0.842 | 0.882 | **0.885** |
| Macro P@1 | 0.437 | 0.767 | 0.819 | 0.830 | 0.875 | 0.875 | **0.890** |

putes the similarity between context words with mention sense, instead of entities, thus achieves the best performance, which also demonstrates the high quality of the mention sense embeddings.

### 6.4.2 Unsupervised Entity Linking

Linking a mention to a specific entity equals to disambiguating mention senses since each candidate entity corresponds to a mention sense. As described in Section 5, we disambiguate senses in two orders: (1) **L2R** (from left to right), and (2) **S2C** (from simple to complex).

We evaluate our unsupervised disambiguation methods on the entire AIDA dataset. To be fair, we choose the state-of-the-art unsupervised methods which are proposed in (Hoffart et al., 2011; Alhelbawy and Gaizauskas, 2014; Cucerzan, 2007; Kulkarni et al., 2009; Masumi Shirakawa and Nishio, 2011) using the same dataset.

Table 5 shows the results. We can see that our two methods outperforms all other methods, while MPME (L2R) is more efficient and easy to apply.

We analyze the results and observe a disambiguation bias to popular senses. For example, there are three mentions in the sentence "*Japan began the defence of their Asian Cup I title with a lucky 2-1 win against Syria in a Group C championship match on Friday*", where the country name *Japan* and *Syria* actually denote the entity of their national football teams, and the football match name *Asian Cup I* has little ambiguity. Compared to the team, the sense of country occurs more frequently and has a dominant prior probability, which greatly affects the disambiguation. By incorporating local similarity and global probability, both the context words (e.g. *defence* or *match*) and the neighbor mentions (e.g. *Asian Cup I*) provide us enough clues to identifying a soccer related mention sense instead of the country.

**Influence of Smoothing Parameter** As mentioned above, a mention sense may possess a dominant prior probability and greatly affect the disambiguation. So we introduce a smoothing parameter $\gamma$ to controls its importance to the overall probability. Figure 3 shows the linking accuracy under different values of $\gamma$ on the dataset of AIDA.

$\gamma = 0$ denotes we don't use any prior knowledge, and $\gamma = 1$ indicates the case without smoothing parameter.

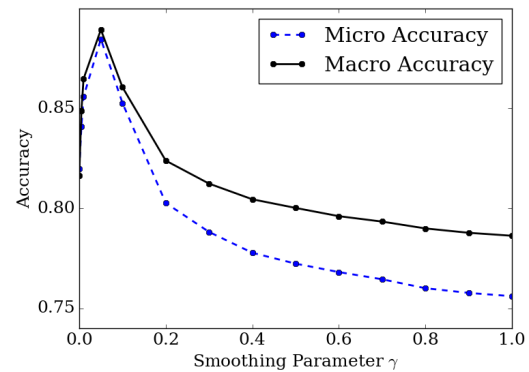

Figure 3: Impact of Smoothing Parameter $\gamma$.

We can see that both micro and macro accuracy decrease a lot if we don't use the parameter ($\gamma = 1$). Only using local and global probabilities for disambiguations ($\gamma = 0$) achieves a comparable performance, and when $\gamma = 0.05$, both accuracy reach their peaks, which is optimal and default value in our experiments.

## 7 Conclusion

In this paper, we propose a novel Multi-Prototype Mention Embedding model that jointly learns word, entity and mention sense embeddings. These mention sense capture both textual context information and knowledge from reference entities, and provide an efficient approach to disambiguate mention sense in text. We conduct a series of experiments to demonstrate that multi-prototype mention embedding improves the quality of both word and entity representations. Using entity linking as a study case, we apply our disambiguation method as well as the multi-prototype mention embeddings on the benchmark dataset, and achieve the state-of-the-art.

In the future, we will improve the scalability of our model and learn multi-prototype embeddings for the mentions without reference entities in a knowledge base, and introduce compositional approaches to model the internal structures of multi-word mentions.

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
