# Peer review of "Bridge Text and Knowledge by Learning Multi-Prototype Entity Mention Embedding"

_ACL 2017 — decision unknown_

[Official Review · Reviewer 1 · rating 3 · confidence 3]
soundness 4 · originality 3 · clarity 3 · impact 3 · substance 4 · appropriateness 5 · meaningful comparison 2 · presentation format Poster

- Strengths:
* Outperforms ALIGN in supervised entity linking task which suggests that the
proposed framework improves representations of text and knowledge that are
learned jointly.
* Direct comparison with closely related approach using very similar input
data.
* Analysis of the smoothing parameter provides useful analysis since impact of
popularity is a persistent issue in entity linking.

- Weaknesses:
* Comparison with ALIGN could be better. ALIGN used content window size 10 vs
this paper's 5, vector dimension of 500 vs this paper's 200. Also its not clear
to me whether N(e_j) includes only entities that link to e_j. The graph is
directed and consists of wikipedia outlinks, but is adjacency defined as it
would be for an undirected graph? For ALIGN, the context of an entity is the
set of entities that link to that entity. If N(e_j) is different, we cannot
tell how much impact this change has on the learned vectors, and this could
contribute to the difference in scores on the entity similarity task. 
* It is sometimes difficult to follow whether "mention" means a string type, or
a particular mention in a particular document. The phrase "mention embedding"
is used, but it appears that embeddings are only learned for mention senses.
* It is difficult to determine the impact of sense disambiguation order without
comparison to other unsupervised entity linking methods. 

- General Discussion:

[Official Review · Reviewer 2 · rating 4 · confidence 4]
soundness 4 · originality 3 · clarity 3 · impact 3 · substance 4 · appropriateness 5 · meaningful comparison 2 · presentation format Poster

This paper addresses the problem of disambiguating/linking textual entity
mentions into a given background knowledge base (in this case, English
Wikipedia).  (Its title and introduction are a little overblown/misleading,
since there is a lot more to bridging text and knowledge than the EDL task, but
EDL is a core part of the overall task nonetheless.)  The method is to perform
this bridging via an intermediate layer of representation, namely mention
senses, thus following two steps: (1) mention to mention sense, and (2) mention
sense to entity.  Various embedding representations are learned for the words,
the mention senses, and the entities, which are then jointly trained to
maximize a single overall objective function that maximizes all three types of
embedding equally.  

Technically the approach is fairly clear and conforms to the current deep
processing fashion and known best practices regarding embeddings; while one can
suggest all kinds of alternatives, it’s not clear they would make a material
difference.  Rather, my comments focus on the basic approach.  It is not
explained, however, exactly why a two-step process, involving the mention
senses, is better than a simple direct one-step mapping from word mentions to
their entities.  (This is the approach of Yamada et al., in what is called here
the ALIGN algorithm.)  Table 2 shows that the two-step MPME (and even its
simplification SPME) do better.  By why, exactly?  What is the exact
difference, and additional information, that the mention senses have compare4ed
to the entities?  To understand, please check if the following is correct (and
perhaps update the paper to make it exactly clear what is going on).  

For entities: their profiles consist of neighboring entities in a relatedness
graph.                    This graph is built (I assume) by looking at word-level
relatedness of
the entity definitions (pages in Wikipedia).  The profiles are (extended
skip-gram-based) embeddings.  

For words: their profiles are the standard distributional semantics approach,
without sense disambiguation.  

For mention senses: their profiles are the standard distributional semantics
approach, but WITH sense disambiguation.  Sense disambiguation is performed
using a sense-based profile (‘language model’) from local context words and
neighboring mentions, as mentioned briefly just before Section 4, but without
details.  This is a problem point in the approach.  How exactly are the senses
created and differentiated?  Who defines how many senses a mention string can
have?  If this is done by looking at the knowledge base, then we get a
bijective mapping between mention senses and entities -– that is, there is
exactly one entity for each mention sense (even if there may be more entities).
 In that case, are the sense collection’s definitional profiles built
starting with entity text as ‘seed words’?                    If so, what
information
is used
at the mention sense level that is NOT used at the entity level?  Just and
exactly the words in the texts that reliably associate with the mention sense,
but that do NOT occur in the equivalent entity webpage in Wikipedia?  How many
such words are there, on average, for a mention sense?                    That is,
how
powerful/necessary is it to keep this extra differentiation information in a
separate space (the mention sense space) as opposed to just loading these
additional words into the Entity space (by adding these words into the
Wikipedia entity pages)?  

If the above understanding is essentially correct, please update Section 5 of
the paper to say so, for (to me) it is the main new information in the paper.  

It is not true, as the paper says in Section 6, that “…this is the first
work to deal with mention ambiguity in the integration of text and knowledge
representations, so there is no exact baselines for comparison”.  The TAC KBP
evaluations for the past two years have hosted EDL tasks, involving eight or
nine systems, all performing exactly this task, albeit against Freebase, which
is considerably larger and more noisy than Wikipedia.  Please see
http://nlp.cs.rpi.edu/kbp/2016/ .  

On a positive note: I really liked the idea of the smoothing parameter in
Section 6.4.2.

Post-response: I have read the authors' responses.  I am not really satisfied
with their reply about the KBP evaluation not being relevant, but that they are
interested in the goodness of the embeddings instead.  In fact, the only way to
evaluate such 'goodness' is through an application.  No-one really cares how
conceptually elegant an embedding is, the question is: does it perform better?

[Official Review · Reviewer 3 · rating 4 · confidence 4]
soundness 4 · originality 3 · clarity 3 · impact 3 · substance 4 · appropriateness 4 · meaningful comparison 2 · presentation format Oral Presentation

- Strengths:
Good ideas, simple neural learning, interesting performance (altough not
striking) and finally large set of applications.

- Weaknesses: amount of novel content. Clarity in some sections. 

The paper presents a neural learning method for entity disambiguation and
linking. It introduces a good idea to integrate entity, mention and sense
modeling within the smame neural language modeling technique. The simple
training procedure connected with the modeling allows to support a large set of
application.

The paper is clear formally, but the discussion is not always at the same level
of the technical ideas.

The empirical evaluation is good although not striking improvements of the
performance are reported. Although it seems an extension of (Yamada et al.,
CoNLL 2016), it adds novel ideas and it is of a releant interest.

The weaker points of the paper are:

- The prose is not always clear. I found Section 3 not as clear. Some details
of Figure 2 are not explained and the terminology is somehow redundant: for
example, why do you refer to the dictionary of mentions? or the dictionary of
entity-mention pairs? are these different from text anchors and types for
annotated text anchors?
- Tha paper is quite close in nature to Yamada et al., 2016) and the authors
should at least outline the differences.

One general observation on the current version is:
The paper tests the Multiple Embedding model against entity
linking/disambiguation tasks. However, word embeddings are not only used to
model such tasks, but also some processes not directly depending on entities of
the KB, e.g. parsing, coreference or semantic role labeling. 
The authors should show that the word embeddings provided by the proposed MPME
method are not weaker wrt to simpler wordspaces in such other semantic tasks,
i.e. those involving directly entity mentions.

I did read the author's response.